# Neuro-Inflammation in Pediatric Traumatic Brain Injury—from Mechanisms to Inflammatory Networks

**DOI:** 10.3390/brainsci9110319

**Published:** 2019-11-09

**Authors:** Erik Fraunberger, Michael J. Esser

**Affiliations:** 1Alberta Children’s Hospital Research Institute, Calgary, AB T3B 6A8, Canada; erik.fraunberger@ucalgary.ca; 2Hotchkiss Brain Institute, University of Calgary, Calgary, AB T2N 4N1, Canada; 3Department of Pediatrics, Cumming School Of Medicine, University of Calgary, Calgary, AB T2N 4N1, Canada

**Keywords:** Pediatric traumatic brain injury, inflammation, secondary injury, network analysis

## Abstract

Compared to traumatic brain injury (TBI) in the adult population, pediatric TBI has received less research attention, despite its potential long-term impact on the lives of many children around the world. After numerous clinical trials and preclinical research studies examining various secondary mechanisms of injury, no definitive treatment has been found for pediatric TBIs of any severity. With the advent of high-throughput and high-resolution molecular biology and imaging techniques, inflammation has become an appealing target, due to its mixed effects on outcome, depending on the time point examined. In this review, we outline key mechanisms of inflammation, the contribution and interactions of the peripheral and CNS-based immune cells, and highlight knowledge gaps pertaining to inflammation in pediatric TBI. We also introduce the application of network analysis to leverage growing multivariate and non-linear inflammation data sets with the goal to gain a more comprehensive view of inflammation and develop prognostic and treatment tools in pediatric TBI.

## 1. Introduction

Traumatic brain injury (TBI) has the potential to produce persistent and intractable medical problems. Referencing the complex nature of the structures damaged following injury, 19th century physicians referred to “molecular disarrangement” as a physiological explanation for the observed negative functional outcomes [1]. We now know the perceived “molecular disarrangement” is the result of a systematic disturbance to homeostatic processes, including cerebral metabolism, blood flow, and synaptic transmission, coupled with immune system activation.

While the sheer complexity of outcomes after central nervous system (CNS) injury in the developed brain is great, injuries to the developing brain present a significant challenge. With ongoing synaptogenesis and myelination laying the groundwork for important neural networks, it is possible that an injury of even mild severity can have an effect on behavior and molecular relationships, such as inflammation, and requires greater understanding of these complex networks and their interactions. The reality is, despite advances in experimental and medical science, all clinical trials for TBI, to date, have failed. Why have our immense amounts of biological data and well-thought out scientific designs yielded no viable results for any age group?

To help explore this question, this review summarizes our current understanding of inflammation in pediatric TBI from the perspective of the central nervous and peripheral immune systems. Moreover, this paper identifies significant knowledge gaps, where understanding of the role of inflammation in pediatric TBI is still incomplete, presenting opportunities for future research endeavors to improve knowledge and outcomes for patients.

While specific sections have been devoted to unique cell types, such as microglia and neutrophils, the relevant literature detailing known cell–cell interactions are included where appropriate, to emphasize the dynamic and interactive nature of the post-TBI immune response. Although this review focuses on inflammation in pediatric TBI, details surrounding what is known about the TBI-induced inflammatory response of adult humans or animals are presented in cases where there is little or no evidence to support a mechanism in the pediatric TBI literature.

In the final section of the review, we outline a novel approach to understanding the interplay among inflammatory factors through the application of graph theory and network analysis. Instead of viewing inflammation from the perspective of single molecules or timepoints, we advocate embracing the inherent dynamism and complexity found when combining observations of multiple mediators over time to grow our understanding and repertoire of prognostic and treatment targets for inflammation in pediatric TBI.

## 2. Overview of TBI

Traumatic brain injury (TBI) is one of the leading causes of death and disability for all age groups in multiple countries around the world [2]. Due to the stage of their brain development, children who sustain a TBI are especially susceptible to acute and chronic cognitive and behavioural changes, including sleep disturbances, impaired attention/memory, emotional lability, and motor/balance issues [3]. 

Traumatic brain injury, as defined by the Centre for Disease Control (CDC), is a disturbance in brain function caused by an external force applied to the head [4], commonly referred to in the literature as the “primary” injury. These external forces can impact the brain in linear and non-linear (ie., rotational) ways to produce physical damage to the brain tissue. While linear forces are known to cause more severe brain injuries including skull fractures, epidural and subdural hematomas, subarachnoid hemorrhage, and cerebral contusions, it is also well established that rotational forces play a key role in all severities of TBI [5]. This is thought to be related to the inherent physical properties of the brain tissue, with a greater resistance to compression (large bulk modulus) and greater susceptibility to deformation from transverse forces (small shear modulus) [6]. Due to the extended, branching structure of white matter and axonal projections, as well as transitional junctions between white matter and gray matter, rotational forces can have a significant effect on structural integrity by producing diffuse axonal injury. The structural damage to the brain tissue and observed symptoms can be used to classify the severity of the injury. Several scales have been proposed to classify the severity of TBIs, including the Mayo Clinic Classification System [7], and the Glasgow Coma Scale (GCS) [8,9], typically in conjunction with structural imaging and clinical symptoms [10] (Table 1). The authors of the Mayo Clinic system found it has a strong ability to classify moderate–severe head injuries across all age groups, compared to other measures, such as GCS. In pediatric patients, however, the GCS has demonstrated value in identifying clinically important head injures [11], although this can be at the expense of high inter-rater variability [12] and limited prognostic value [13]. It is also important to note most TBIs will realistically have some combination of focal and diffuse patterns of injury, considering the complex biomechanical forces at work. For severe TBI, focal injuries will be present alongside diffuse axonal injury and potentially lead to an impaired cognitive and behavioural phenotype. In contrast, mild injuries are often defined on the basis of their lack of focal injury (hemorrhages or lesions), and have shown alterations in white matter tracts in clinical studies [14,15,16] and in animal models [17,18]. 

As a consequence of brain tissue damage, constituent cells and surrounding vasculature undergo structural and molecular changes, collectively known as “secondary” injury. These changes include excitotoxicity, mitochondrial/metabolic dysfunction, increased oxidative stress, weakened blood–brain barrier (BBB) integrity, cytoskeletal breakdown and protein aggregation, cerebral blood flow dysregulation, edema (vasogenic and cytotoxic), and inflammation. While these mechanisms of secondary injury are often described and targeted separately in preclinical and clinical studies, it can be argued that many, if not all, work in conjunction with immune cells in the brain and periphery to induce and propagate the inflammatory response after all severities of TBI. This is important, as chronic inflammation following a brain injury is thought to be biologically linked to increased risk of neurodegeneration later in life [19]. Figure 1 describes some of the key mechanisms related to inflammation after TBI.

## 3. Pediatric TBI—Different Than Adults

Described previously by Giza et al. [20], the developing brain reacts differently to injury than the adult brain. Key physiological differences in the brains of children include ongoing myelination and synaptogenesis [21], generally greater cerebral blood flow and oxygen consumption until adolescence [22,23], increased brain water content [24], and less elastic brain parenchyma [25]. Gross anatomical differences, notably a larger head relative to body size [26] and lower center of gravity [27], also increase pediatric susceptibility and influence outcomes related to TBI (Figure 2).

In pre-clinical studies, pediatric TBI has been modeled using a variety of animals, type of TBI, age at time of injury, post-injury analysis time points, behavioural tests, and molecular markers, in an effort to more clearly define how the developing brain responds to injury. As described in a detailed review of pediatric TBI models by Kochanek et al., rodents aged P7–P21 are typically used to model pediatric TBI up to the age of a human toddler [28]. This age range encompasses the perinatal and weanling stages of the rodent life cycle. To model injuries related to human children older than a toddler and undergoing puberty, rodents from P30 up to sexual maturation, typically between P32–P34 in females and P42–P50 in males, are often used [29]. However, these age ranges and correlations with humans are approximate, as equating rodent and human development depends on the life stage of the animal, since the speed of development is significantly faster in rodents compared to humans [30]. Although the molecular and behavioural changes occurring in the rodent brain can be somewhat correlated with human biology, piglets are growing in popularity due to their brains having a similar neurodevelopmental trajectory, white:grey matter ratio, and gyrencephalic anatomy, compared to humans [31]. In addition to the choice of animal model, a caveat for all researchers to consider is that laboratory animals are raised in a relatively sterile environment, which may impact immune system programming and function . Considering humans are exposed to a wide variety of pathogens in the environment, and these pathogens serve as a “learning” platform for the immune system, it is conceivable the immune response experienced by an injured child may differ from that experienced by a lab animal exposed to minimal pathogens. 

While delving into each of the experimental TBI models, such as a controlled cortical impact or weight drop, is beyond the scope of this review, it is important to note these models restrict craniocervical movement (e.g., controlled cortical impact, fluid percussion injury) or have been used to produce moderate to severe injury (e.g., modified Feeney weight drop). In contrast, comparably less work has been done using models with a high face validity for the causes and mechanisms of mild pediatric TBI. Although it has been rightly argued that current models have high reproducibility, since they rigorously control impact variables such as depth and velocity, future work should also incorporate rotational mechanisms of injury. While this may produce greater behavioural heterogeneity, as our group has documented [32,33], this more accurately reflects clinical realities and heterogeneity between individuals across the age spectrum in pediatric TBI.

## 4. Inflammatory Response to TBI

For the sake of simplicity, the inflammatory response following a TBI, , can be subdivided into the central (i.e. CNS-based) and peripheral responses. However, this does not discount the substantial interactions between the neuroimmune cells and peripheral immune cells. Further, the degree of inflammatory response is associated with the severity of the injury, as more severe injuries generally elicit a greater and more robust immune response, as measured by cytokines and immune cell activation. Milder forms of TBI may produce very little, if any, tangible evidence of inflammation. Below, we provide an overview of cell-specific mechanisms that contribute to the post-TBI inflammatory response. Table 2 provides an overview of CNS-based mechanisms of secondary injury and how they can promote the inflammatory response to TBI. 

Within the central nervous system, the primary mediators of TBI-induced inflammation are astrocytes and microglia. Depending on the severity of injury, reactive gliosis may occur where astrocytes and microglia alter their structure and function along an activation continuum in response to damage. Originally conceived as a polarizing phenomenon, with “neurotoxic”/”pro-inflammatory” and “neuroprotective”/”anti-inflammatory” phenotypes predominating at various timepoints after injury, this concept has since been widely recognized as an over-simplification. Using contemporary molecular techniques to measure metabolism, and various “-omics” strategies including RNA-seq, it is now understood that glial cell activation exists as a spectrum, with significant overlap in expression between traditional “pro” and “anti” inflammatory markers in astrocytes (see [34]) and microglia (see [35]). 

Considering most TBIs are classified as closed-head, and this form of injury is considered sterile, typical ligands of pattern recognition receptors (PRRs) from infectious sources, such as lipopolysaccharide and dsRNA, are not likely to be the primary immunogenic stimuli. Rather, intracellular proteins, serving as Damage-associated molecular pattern (DAMPs), such as S100β, heat-shock proteins, mitochondrial DNA [36], and HMGB1, can activate surface PRRs, such as Toll-like receptor 2 (TLR2) and TLR4, and intracellular PRRs, such as TLR9 and the cGAS-STING pathway, to elicit an immune response. Despite considerable evolutionary homology, caution must be taken when comparing rodent to human studies, as TLR expression is known to be cell-dependent and variable between human and rodent brains [37]. For example, while rodents and humans show a similar expression of TLR2 on microglia, astrocytes in the rodent brain show TLR1-9 expression, and human astrocytes lack expression of TLR6-8 [38]. 

### 4.1. Astrocytes

As the primary cell type found to be activated following TBI, astrocytes play a key role in the initial immune response. TBI-induced activation of astrocytes is thought to be similar to neurons, where stretch-sensitive ion channels (e.g., NMDA receptors and Big Potassium channels) on the surface of the astrocytes become deformed and allow for an extensive Na^+^ and Ca^2+^ influx and K^+^ efflux [59]. Concomitant ATP release, which initiates autocrine and paracrine Ca^2+^ influx, induces further ATP, glutamate, and S100β release (via MAPK signaling [60]), microglia recruitment to the site of injury, and the subsequent activation of both astrocytes and microglia [61]. Microglia within the immediate vicinity of the injury also become activated and can secrete IL-1α, TNFα, and C1q to induce the phenotypic conversion of astrocytes to an A1 or “neurotoxic” profile [54], illustrating the reciprocal signaling and relationship between these glial cells. After injury, astrocytes have been found to differentially express cell activation markers, including complement component 3 (C3), CD109, Stat3, Tgm1, and CD14 (see [62] for a detailed review). However, all of these markers have been characterized in adult CNS, and more work is needed to determine if this holds true in the pediatric brain and through the course of development. Surface expression of receptors for advanced glycation end products (RAGE) and TLRs on astrocytes also provide an additional mechanism by which DAMPs can activate NF-κB signaling, enhancing the transcription of pro-inflammatory cytokines, such as TNFα and IL-1α, and pro-inflammatory enzymes, such as cyclooxygenase-2 (COX-2) and matrix metalloproteinase 9 (MMP9) [63].

Due to the connections of astrocytes in a functional syncytium via gap junctions comprised of connexins [64], Ca^2+^ oscillations in response to mechanical and chemical (i.e., glutamate and ATP) stimuli can indirectly induce neuronal activity at, and distant to, the site of injury, exacerbating the excitotoxic environment. Within the astrocyte, the increase in intracellular Ca^2+^ can then act to: (1) directly affect the transcription of activation markers, like GFAP and vimentin, via Ca^2+^ sensitive kinases and phosphatases [65,66,67], and (2) increase the production of ROS, via potentiation of the citric acid (TCA) cycle and electron transport chain (ETC) [68], and, subsequently, production of inflammatory proteins, such as IL-1β and IL-6, by activating the NLRP3 inflammasome [69]. In milder forms of TBI, astrocytes respond to damage by undergoing reactive hypertrophy within 24 h of injury through the upregulation of GFAP and vimentin structural protein expression [59]. A similar phenomenon has been observed in humans following TBI with reactive gliosis, from the perspective of GFAP expression, occurring between 24 and 72 h post-injury [70]. In the same population, protein levels of excitatory amino acid transporter 2 (EAAT2) were 50% lower than in controls and remained depressed up to at least 7 days post-injury. At the most extreme end of the severity spectrum, where neurons, glial cells, and neighbouring blood vessels are damaged and DAMPs are leaking into the extracellular space, activated astrocytes seal off the damaged tissue and contain inflammatory stimuli by creating a glial scar. While the upregulation of ephrins and ephrin receptors on astrocytes and surrounding cells, notably EphA4 [71] and EphB2 [72], and astrocytic secretion of extracellular matrix substrates, such as chondroitin sulfate proteoglycans (CSPGs) and collagen, directly inhibit axon growth and regeneration, the glial scar is actually considered an “anti-inflammatory” response to CNS injury [73]. However, all of the above observations are from adult rodent and human brains, typically male, and it has yet to be determined whether the response is different in female and/or pediatric brains. 

### 4.2. Microglia

Microglia play important roles in damage response and control through constant communication with neurons, astrocytes, and blood vessels [74]. At rest, microglia maintain a ramified morphology, with processes that continuously survey their surrounding environment. To survey synapses and stabilize dendritic spines, microglia make use of an ATP-dependent process. For synaptic pruning during postnatal development, microglia eliminate less active synapses using the complement system, notably via C1q, CXCL1, and C3 signaling [75]. Although these functions appear to be ubiquitous and consistent throughout the CNS, microglia possess regional morphological and functional heterogeneity throughout the brain that changes with age [76]. For example, cerebellar microglia, at least in rodents, are noted to possess a less ramified morphology and greater somal motility compared to their cortical counterparts [77]. 

The normal glial homeostatic mechanisms described above can be disrupted by a TBI, with activation peaking around 7 days post-injury. If an injury occurs, damaged neurons and astrocytes release adenosine and phosphorylated derivatives such as ATP, which bind to P2Y receptor subtypes [51], notably P2Y12 [78], on the surface of the microglial processes to attract them to the injury site. Once microglia arrive and encounter successive DAMPs, they adopt an amoeboid morphology to facilitate phagocytosis of cellular debris. However, microglial morphology can be heterogenous depending on the injury location and type. For example, when in contact with the glia limitans following a meningeal compression injury, microglia are able to adjust their morphology and surround individual astrocytes with processes to stabilize the glia limitans [61]. More generally, within the first 24 h following TBI, microglia appear to adopt a phenotype where they secrete cytokines and chemokines, including IL-1β, TNF-α, IL-12 and IL-6, and increase reactive oxygen/nitrogen species (ROS/RNS) production through upregulated NADPH oxidase and cyclooxygenase 2 (COX-2) activity [79]. On the surface of the microglia, DAMP interactions with TLRs upregulate various clusters of differentiation (CD) receptors, major histocompatibility complex II (MHC-II), cytokines, chemokines, inducible nitric oxide synthase (iNOS), arginase-1 (Arg-1), triggering receptors expressed on myeloid cells 2 (TREM2) and the mannose receptor (CD206), to name a few. Despite the consistent use of certain microglial activation markers in preclinical research, including Arg-1 and chitinase-like 3 (Ym1) [80], caution is warranted, as these animal studies have yet to be conclusively validated in humans. More work is needed to understand if the expression of activation markers differs in the pediatric brain, and if all markers used preclinically can translate to the developing human brain. 

### 4.3. Pericytes

Outside of astrocytes and microglia, pericytes play an important role in the inflammatory response following TBI. As a key component of the neurovascular unit, pericytes help regulate capillary blood flow and immune cell entry into the CNS. Pericytes can also act as immune cells by responding to DAMPs and cytokines secreted by nearby glial cells and by expressing inflammatory factors, including various chemokines (CXCL10, CCL20, CXCL8, CCL2, etc.), chemoattractant proteins, Il-1β, IL-6, and IL-8 [81].

Using the focal Marmarou weight drop model of TBI on 6–8-week-old rats, Dore–Duffy et al. noted that 40% of pericytes near the site of injury migrated away from the vasculature into the ipsilateral brain parenchyma within 1–2 h of injury [82]. Given the age range used in this study, further work is needed to determine if any age-specific effects can be observed in pericytes following TBI. However, this finding has been replicated in an adult mouse CCI model, where there was an acute loss of pericytes up to 12 h post-injury, followed by a spike in reactive pericytes around the pericontusional area between 3 and 5 days post-injury [83]. Although not clearly delineated, hypoxic endothelial cells can release angiogenic factors, such as hypoxia-inducible factor 1, that signal for the pericyte to detach from the vessel wall and degrade the extracellular matrix in preparation for blood vessel growth [84]. Acute loss of pericytes around the endothelium can also enhance BBB permeability by altering the spatial expression of astrocyte aquaporin-4 (AQP4) away from endfeet near blood vessels towards the soma [85]. Given the previous work in focal, adult rodent TBI studies demonstrating a decrease in AQP4 at the lesion site at 1–24 h post-injury [86,87], followed by an increase in AQP4 and intracellular edema at 3 h post-injury in the perilesional area [87], changes in AQP4 expression may be a compensatory mechanism to reduce vasogenic edema and indirectly induce cytotoxic edema [88]. In conjunction with the timing of pericyte loss, it is possible the compensation for vasogenic edema may be a consequence of pericyte detachment from the basement membrane and an altered polarization of astrocyte endfeet. Overall, acute loss of pericytes after injury is thought to increase BBB permeability to peripheral macromolecules, immune cells, and water, potentially promoting an inflammatory response and edema formation.

In a biphasic manner, pericytes return between 3–5 days post-injury in the adult mouse, alongside normalized AQP4 expression, even though BBB permeability is still compromised [86]. In juvenile animals, however, this increase in astrocytic AQP4 expression coincides with edema resolution [89] but not with reduced BBB permeability, as the BBB was found to remain somewhat permeable to IgG between 3 and 7 days post-CCI [90]. As noted by Simon et al., in their review of post-TBI neuroinflammation [91], most of the peripheral immune response to TBI occurs between 24 h and 7 days post-injury, reinforcing the idea that increased BBB permeability following TBI, partially due to pericyte loss, plays a large role in immune response to injury.

### 4.4. Mast Cells

Although they are derived from myeloid stem cells in the bone marrow, mast cells are another cell line that is thought to play an important role in the inflammatory response following TBI. Within the CNS, mast cells have been found in the dura mater near blood vessels and sensory nerve fibers [92], the choroid plexus, thalamus, hypothalamus, pituitary stalk, and pineal gland [93]. Mast cells have been noted to secrete various biologically active proteins, such as histamine, heparin, proteolytic enzymes, phospholipases, somatostatin, endorphins, dopamine, and serotonin, at least in rodents [94]. During a period of injury or infection, mast cells have been found to induce microglial activation in both in vitro [95] and in vivo [96,97] rodent models. After a focal TBI in adult Sprague–Dawley rats, meningeal mast cell degranulation has been shown to increase histamine receptor 3 (H_3_) binding density, and decrease mRNA levels of H_1_, H_2_ and superoxide dismutase (SOD) in the ipsilateral thalamus [98]. Closed-head models of diffuse injury in adult mice and rats have also shown meningeal mast cell degranulation, increasing cortical histamine levels [99] and contributing to chronic latent sensitization to pain [100], with degranulation persisting for up to 30 days post-injury [101]. However, Moretti et al. found that although mast cells can activate microglia following TBI in a P7 mouse, inhibition of degranulation with cromoglycate or knockout of mast cells did not affect neural cell loss [102]. Thus, it is still difficult to interpret the contribution of mast cells to the pathophysiology of pediatric TBI at this time, as there are still too few mechanistic studies in age appropriate animal models. 

### 4.5. Peripheral Immune Response to TBI

Peripheral immune cells constitute key players in the inflammatory response following TBI in the pediatric population. Previous work has documented some differences in peripheral immune cell prevalence in the pediatric population up to 18 years of age. For example, lymphocytes gradually increase in children from 6 months to 18 years of age, while neutrophils and eosinophils decrease between the ages of 14 and 15–18 years of age [103]. This has important implications for the immune response to TBI, as the age of the child can affect the composition of immune cells responding to injury and the subsequent inflammation.

Broadly speaking, the peripheral immune response can be subdivided into innate and adaptive responses, with the innate taking place within the first 2 days following injury while the adaptive becomes more prominent beyond 48 h [104]. As mentioned previously, this timeline has yet to be rigorously evaluated in the human pediatric population or in adolescent animals, so inferences and extrapolation of results should be interpreted cautiously. Given the BBB disruption following injury, especially in cases of focal TBI, where the BBB can become permeable within 3 minutes of injury [105], peripheral immune cells are recruited by DAMPs and chemokines secreted by activated astrocytes, microglia, pericytes, and other neuroimmune cells. The secretion of substance P from sensory neurons surrounding the damaged vasculature can interact with neurokinin 1 (NK1) receptors on vascular endothelial cells and lead to the extravasation of material, including albumin [106], and potentiate the activation of native immune cells and secretion of DAMPs [107]. Once leukocytes have been attracted to the BBB, numerous adhesion receptors expressed on the surface of the endothelial cells, such as ICAM-1, VCAM-1, and P-selectin, interact with leukocyte-based adhesion ligands, including integrin alpha L-beta 2, integrin alpha-4, and chemokine receptors, to initiate the attachment, rolling, and extravasation into the brain. 

As the predominant circulating leukocyte and the first cell type to respond, in part due to the secretion of CXC family chemokines and various cytokines through the BBB [108], neutrophils can be found in the brain tissue after focal TBI as early as 2 h after injury. Their numbers peak between 24 and 48 h, and decrease to baseline levels around 7 days post-injury. Neutrophils can release cytotoxic factors, including cytokines, neutrophil elastase (degrades extracellular matrix), and reactive oxygen species, which may exacerbate injury. Work done in P21 rats using a CCI model of injury has shown neutrophil elastase to be a major contributor to vasogenic edema and spatial memory impairments after injury [109]. A caveat of these results and timeline is that it has been generated through studies largely involving severe, focal injury models, such as CCI, or weight drop with the skull removed. In a moderate, diffuse weight drop injury in adult [110,111] and P17 [112,113] Sprague–Dawley rats, neutrophil infiltration was undetectable, even though BBB permeability was not disrupted. Using a similar weight drop model, Trahanas et al. noted monocytes comprise approximately 46% of infiltrating leukocytes, supporting the previous findings of lack of neutrophil presence after a diffuse TBI [114]. However, it remains to be seen whether neutrophil infiltration is also a significant mechanism of inflammation following milder forms of injury, or in injuries without a focal component. As mentioned above, neutrophil counts may decrease between the ages of 14 and 15, raising the possibility of a differential age-related neutrophil response to TBI in children, but this has yet to be studied in depth. 

Facilitating neutrophil infiltration, monocytes are attracted by a host of chemokines, including CCL2, CCL3, CCL5, CCL7, CCL8, CCL13, CCL17, and CCL22 [115], and have been documented in lesioned brain tissue and the surrounding perivascular spaces within 24–48 h after severe TBI in humans [116]. Phenotypically similar to microglia in terms of functional characteristics and receptor expression, monocytes become activated in the CNS upon exposure to DAMPs, including purines (i.e., ATP, ADP) and cytokines (notably colony stimulating factors). Once activated within the CNS, monocytes/macrophages can act as a double-edged sword by producing neurotrophic factors (i.e., BDNF, NGF), or chemokines to attract neutrophils (i.e., CXCL1) and additional monocytes (i.e., RANTES), depending on the inflammatory environment. There is recent evidence of neuroprotection when peripheral macrophages were found to dampen the microglial response following spinal cord injury [117]. Given the complex nature of the immune response to CNS injury and documented heterogeneity of microglia, it is likely many of these responses occur simultaneously by differing subsets of macrophages in the lesioned, perilesional, and contralateral areas, in the case of a focal TBI, or in different regions throughout the brain in the case of a diffuse injury. However, while this heterogeneity has been demonstrated in the mouse spinal cord [118], this remains to be explored in both pediatric TBI models and in females, despite recent evidence of sex differences in myeloid cells contributing to post-TBI inflammation in adult mice [119]. Future studies will also need to consider the prevalence of different monocyte subsets in humans versus mice, and the age differences in monocyte chemotaxis. For example, the “non-classical” subset of monocytes, widely understood to patrol tissue vasculature and respond to damage by recruiting neutrophils [120], comprise 40% of mouse monocytes and only 5–10% of human monocytes [121,122]. Children under 10 years of age have been shown to have reduced monocyte chemotaxis compared to adults [123,124], potentially reducing the contribution of monocytes to the post-TBI immune response in this age group. However, this remains to be thoroughly investigated. 

Moving beyond 48 h post-injury, the adaptive immune response becomes more prominent. Monocytes, along with dendritic cells, act as antigen presenting cells (APCs) to activate T cells in the peripheral circulation. When brain cells are damaged following a TBI, integral proteins, including myelin basic protein (MBP) [125], GFAP [126], and S100β [127], leak into the circulation and surrounding tissue and are taken up by APCs. Once the APCs enter the lymphatic system via the meningeal lymphatic vessels [128] and the recently characterized glymphatic system [129], they present these and other antigens to immature CD4^+^ and CD8^+^ T cells in the thymus and B cells in the lymph nodes and spleen, where they can activate and propagate the immune response to TBI. However, in other studies, T cell infiltration, namely T cells specific to MBP, can be neuroprotective by reducing retinal ganglion cell loss after axotomy in the adult rat [130]. Similar results would need to be shown in TBI models of any age before similar conclusions could be drawn.

In severe, focal TBI in adult mice, 50% thymocyte loss was observed at 3 days post-injury, blood monocyte counts were reduced by 3.5x at 24 h post-injury, and levels of IL-12 were markedly decreased compared to sham [131]. This immunosuppressed state, at least with respect to levels of peripheral lymphocytes, has also been demonstrated in adult humans, with a suppression of levels of CD4^+^ and CD8^+^ T cells at 4 days post-injury, recovering by 7 days [132]. However, this response may not hold true in children, as the adaptive immune response to severe TBI in pediatric patients has been documented to produce a greater number of leukocytes and cytotoxic T cells (CD4^+^ and CD8^+^) 24 h post-injury, with T cells demonstrating decreased autoreactivity to neuronal antigens in vitro [133]. Similar to animal studies, however, the immune response markedly decreases at 3–5 days post-injury relative to control levels, a finding that has been associated with an increased risk of nosocomial infections following hospitalization for TBI [134,135]. 

Research on the B cell response to TBI in any age group or severity is comparably limited. As such, our understanding of B cell involvement in TBI is from the perspective of autoantibody production to antigens such as MBP and GFAP [136,137]. Work done in children (7–16 years) with chronic post-traumatic headache secondary to mild TBI demonstrated the presence of autoantibodies to the GluR1 and NR2A subunits of the AMPA and NMDA receptors, respectively, up to one year post-injury [138]. Future research into the adaptive immune response to pediatric TBIs of any severity, is needed. For a comprehensive review on the autoantibody response in TBI, see Needham et al [139].

### 4.6. Neurogenic Inflammation After TBI

The least researched of any mechanism of post-injury inflammation, especially in the pediatric population, is neurogenic inflammation. This form of sterile inflammatory response can potentiate classical inflammation (described above) and potentially contribute to post-injury chronic headache and pain [140]. Within the CNS, neurogenic inflammation occurs following mechanical stimulation of transient receptor potential family (TRP) receptors, namely TRPV1 and TRP1A type channels, on sensory neurons associated with vasculature [141]. Activation of these channels leads to Na^+^ and Ca^2+^ influx, facilitating the release of neuropeptides such as Substance P (SP) and calcitonin gene-related peptide (CGRP), which in turn, can act as potent vasodilators and increase vascular permeability [142,143,144]. SP is known to potentiate classical inflammation by interacting with NK1 receptors, present on endothelial cells, astrocytes, microglia, and peripheral immune cells [145,146]. Upon binding to the NK1 receptor, SP induces vasodilation and increases BBB permeability, increases leukocyte adhesion molecule expression on endothelial cells [147,148], increases peripheral immune cell extravasation into the CNS [149], exacerbates excitotoxicity via mast cell degranulation (see above) [150], and increases expression of inflammatory mediators including cytokines (TNF-α, IL-1β, and IL-6), ROS/RNS, and metalloproteinases from microglia, astrocytes, and peripheral immune cells [151,152,153]. In turn, many of these effects can reinforce the neurogenic response through increases in pro-inflammatory molecules, such as prostaglandins [154], which serve to boost the secretion of SP. To the best of our knowledge there have been no studies examining the scope of neurogenic inflammation in the pediatric population or in models of pediatric TBI. Previous research has demonstrated a reduced expression of NK1 receptors and reduced neurogenic inflammatory response in P16 pups compared to adult rats [155], implying the neurogenic inflammation observed in older rodents may not be replicated in younger animals. Nonetheless, numerous adult rodent studies have illustrated a positive association between injury severity and SP release, as well as therapeutic benefit, in the form of reduced BBB permeability and vasogenic edema, following NK1 receptor antagonism [156] or neuropeptide depletion with capsaicin [157]. Given the negative correlation between circulating plasma levels of SP and 30-day mortality following severe TBI in human adults [158], and the high incidence of post-traumatic migraines [159] and non-headache pain [160] in children, investigation of neurogenic inflammation in the pediatric TBI population represents an intriguing future direction for the field. Since SP is known to mediate the development of pain sensitization [161,162] and CGRP is strongly associated with migraines [163], it is possible these two molecules may play a role in the development of post-injury, non-headache pain and migraines in the pediatric population, but this requires deeper investigation.

## 5. Moving Beyond Singular Markers—Taking an Interactive & Dynamic Network Approach to Inflammation

As detailed above, it is well established that immune cells and their various responses constitute a complex and dynamic biological system. Until recently, studies on TBI and inflammation have largely focused on singular, or a handful of, inflammatory mediators at discrete time points after an injury at a single age. Through this approach, great strides have been made in characterizing downstream pathways, including the NF-κB [164] and inflammasome [165] pathways, and the myriad of molecules produced in order to create the inflammatory response. However, viewing these molecules as discrete and individual mediators is not consistent with the principles of systems biology. A new and powerful approach is to leverage our understanding of these signaling pathways and view the inflammatory response to TBI as a set of interacting inflammatory mediators. Taking a step back from individual pathways, this systems biology approach can be applied temporally in the same population or, in the case of animals, the same TBI model over a spectrum of ages to greatly enhance our understanding of the pattern of inflammation after TBI. Using this approach will reduce our reliance on interpolating between studies and hypothesizing about the inflammatory response based on a composite of static studies.

Adopting a systems biology mindset will facilitate the conceptual shift from individual agents to one where each cellular and molecular entity is part of an interconnected, dynamic, and fluid network. Instead of examining the details of each signaling pathway ascribed to a particular inflammatory molecule, each constituent is examined from a broader perspective in terms of its relationships with other members of the network. Using contemporary “-omics” and high-throughput approaches to measuring metabolites (ex. Q-TOF mass spectrometry), proteins (ex. multiplex ELISA), and gene transcripts (ex. RNA-seq), the absolute levels of inflammatory molecules can be quantified and correlated with one another to produce a more complete picture of post-TBI inflammation.

### Biological Networks and Relevance to Inflammation after TBI

The large volumes of biological data being generated can be visualized as networks through the application of graph theory. While the mathematical tenets of graph theory are beyond the scope of this review, we will briefly discuss the network properties, their relevance to biology, and how they can be constructed in terms of inflammation. 

Inflammation networks are typically visualized as nodes, representing individual inflammatory molecules, connected to each other using a pairwise relationship metric (represented as a line or “edge”), such as a correlation coefficient. The power of the networks lies in their ability to temporally and spatially quantify the relationships between each of the nodes following a TBI. This gives a new dimension to our empirical and intuitive understanding of inflammation as a dynamic and interactive process involving the synergistic [166], additive [167], or antagonistic [168] effects of different molecules after injury, over time, and between regions. This is important, as many cytokines, chemokines and growth factors are known to act in multimodal ways, depending on the cellular context and insult mechanism (i.e., sterile vs. infection) and can also act in non-inflammatory ways, including as pro-survival and growth factors [169]. 

Imperative to the implementation of this approach is the understanding of what these networks represent and how they are constructed. Typically, relationships are visualized after importing known associations from a database such as STRING [170]. Following the import of a set of associations previously published in literature, networks can be constructed by users overlaying their own observations, such as the magnitude and direction of a correlation coefficient between two nodes. At this point, the network represents how each inflammatory mediator changes in relation to its neighbours. In the context of TBI, an examination of associations can focus on comparing the magnitude and direction of change between two inflammatory mediators or the overall pattern of inflammatory relationships (i.e., changes in network metrics and clustering) in control and injured conditions. 

When creating these networks using correlations between each inflammatory mediator and analyzing their interactions, it is important to remember the reflexivity and non-linearity embedded in such a complex system [171]. When using feedback loops with neural circuits [172] and, in the case of inflammatory molecules, binding to multiple receptors possessing different binding characteristics [173], observed effects can be highly variable and do not change in easily predictable ways. Most relationships or correlations established between cytokines in a network do not reflect this as they are done using correlation coefficients that assume a linear, monotonic relationship. This is demonstrably false, with a notable example being IL-6 displaying a bimodal distribution of expression following exposure to LPS in mice [174]. Instead of using data transformation to fit our parametric, linear modeling techniques like a Procrustean bed, we suggest it is worth exploring non-linear patterns of correlations using non-parametric coefficients that are less susceptible to outliers, such as Kendall’s tau or Hoeffding’s D [175]. It is important to make use of these relationship metrics in the context of large sample sizes to allow for the true nature of correlations to become evident. Unfortunately, due to constraints on animal studies where sample sizes are typically less than that of human studies, any subsequent network analyses must be interpreted with caution and reproduced before any firm conclusions can be drawn.

Once a network has been constructed, analyzing the relationships or the network connectivity provides us with a deeper understanding of the individual’s response (as a whole) to a brain injury. Further, with the right mathematical tools and rigorous studies, there is the potential for the data to be used as a prognostic marker or guide for targeted therapeutic strategies. Researchers within the field, specifically Scardoni [176,177] and Jalili [178], have provided biologically-relevant interpretations of mathematical entities calculated during network analysis. In using network measurements, it is hoped that we can begin to elucidate the emergent properties of the immune system [179], and its response to TBI, which seems improbable through studying each cytokine alone [180]. 

With our growing understanding of how the inflammatory process can be upregulated or dampened, there is also a strong argument in favor of constructing networks of chemokines and cytokines at multiple time points post-injury. Creating such dynamic networks, as has been argued by others in the field [181], simultaneously broadens our knowledge base of the inflammatory cascade post-TBI and may allow us to identify significant groups of cytokines that are clinically relevant for prognostic or treatment purposes. This also opens up the possibility of constructing inflammatory networks unique to each individual and to track the progression of their inflammation over time after injury. 

A recent application of network analysis to cytokine networks after TBI can be found in Rowland et al.’s analysis of the acute inflammatory response in adult TBI patients [182]. Using a multiplex ELISA, the authors were able to quantify cytokine levels in the blood of TBI patients, calculate correlations between each cytokine, and use hierarchical clustering to reveal novel groupings of cytokines after TBI in adults. Moreover, the authors take a broader view of the inflammatory networks and demonstrate an altered pattern of connectivity and clustering between several cytokines in TBI patients compared to controls. While it is premature to infer these results to a pediatric population, the development of a methodology to identify groups of altered cytokines, chemokines, and other inflammatory mediators after TBI can provide a path for the development of novel, multimodal therapies. 

## 6. Conclusions and Future Directions for Research and Therapy

Our attempts to construct a timeline of inflammation after TBI are largely based on (1) the adult age group in both humans and animals; (2) focal, moderate–severe injuries, largely neglecting rotational mechanisms of injury; (3) the male sex; and (4) singular time points after injury [91,183]. Leveraging our knowledge of the signaling pathways underlying inflammation and combining them with (1) a multiplex approach measuring multiple inflammatory molecules; (2) in models incorporating rotational mechanisms of injury; (3) using both sexes; (4) at multiple time points in the under-researched pediatric age group is the next step for the field.

Why is such a network approach necessary? The answer lies in the reality of 500 completed, suspended, withdrawn, or terminated clinical trials for traumatic brain injury, of which 127 have been conducted in the pediatric population. As was recently and aptly pointed out by Appavu et al., every trial has failed to produce high-quality evidence for the treatment of pediatric TBI [184]. A common theme across these trials is the targeting of a singular mechanism of secondary injury and use of a small number of biomarkers to evaluate success. Although we intuitively understand the complexity inherent in the body during both homeostatic and injured states, many of our clinical trials directed at inflammation following TBI have taken a reductionist approach and have been based on research targeting one inflammatory mechanism. In the future, instead of targeting a single inflammatory molecule and its associated receptors, a more diverse response towards inflammation, such as targeting a range of mechanisms or different mechanisms at different time points, is needed. For example, if a network analysis were to identify a group of several cytokines consistently altered after a TBI in the pediatric population, it would be possible to model the antagonization of one or multiple cytokines, and its effect on network coherence and connectivity. This would then be re-evaulated, along with the clinical trajectory, and adjusted as needed. As shown in Figure 3, this methodology has the potential to generate novel therapies for TBI and significantly advance our understanding of inflammation in the pediatric population. Future work will be needed to discover which network characteristics, such as increased clustering or connectivity, provide the best way to target inflammation. However, we believe such an effort will produce meaningful methods of analyzing complex biological data, refine our preclinical models of pediatric TBI, and improve the bidirectional feedback between bench and bedside to the benefit of those affected by pediatric TBI.

## Figures and Tables

**Figure 1 brainsci-09-00319-f001:**
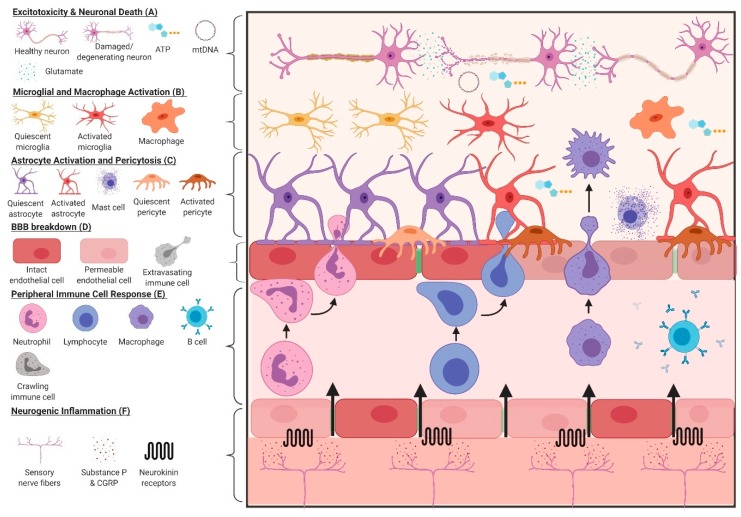
Key mechanisms of inflammation following TBI. (**A**) Following a TBI, an increased influx of Ca^2+^ and Na^+^ into neurons facilitates excessive glutamate release into the synaptic cleft and consumption of ATP stores by Na/K ATPases to restore membrane potential. Increased intracellular Ca^2+^ can overwhelm the buffering capacity of the mitochondria and cause membrane depolarization, leading to organelle fragmentation and the impairment of oxidative phosphorylation. If the disruption is severe, cellular death can occur and the leakage of DAMPs into the extracellular space can elicit an immune response. (**B**) Detection of products from neuronal damage, including to myelinated tracts, can activate surrounding microglia via pattern recognition, purinergic, and glutamate receptors. Cells use these chemotactic signals to migrate towards the site of injury and launch an inflammatory response, including the secretion of cytokines and gliotransmitters to alert astrocytes. (**C**) Closer to the blood–brain barrier (BBB), activated astrocytes and pericytes react to damage and extracellular DAMPs by reducing the coherence of the BBB through the retraction of endfeet and migration away from the vasculature. Mast cells on the abluminal side of the BBB can degranulate in response to TBI, releasing extracellular matrix (ECM)-degrading enzymes, neurotransmitters and histamine. (**D**) Increased expression of adhesion molecules on endothelial cells following TBI, such as ICAM-1 and VCAM-1, increase the attachment and extravasation of peripheral immune cells into the brain. (**E**) BBB breakdown and leakage of DAMPs into the peripheral circulation act as chemoattractants for leukocytes. Following TBI, neutrophils, T-cells, and monocytes have been documented to penetrate the BBB and perpetuate the immune response. B-cells can produce autoantibodies to cerebral antigens, such as myelin basic protein (MBP) and glial fibrillary acidic protein (GFAP), propagating inflammation. (**F**) Mechanical stimulation of transient receptor potential (TRP) channels on sensory nerve fibers can release substance P (SP) and calcitonin gene-related peptide (CGRP), increasing the permeability of the endothelium to immune cells and DAMPs, and facilitating vasogenic edema. Created with Biorender.com.

**Figure 2 brainsci-09-00319-f002:**
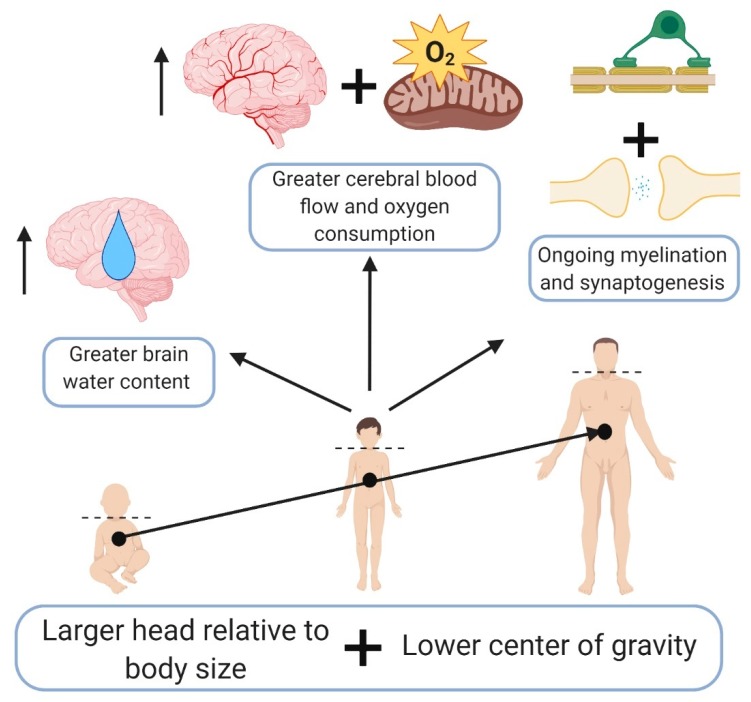
Key anatomical and physiological differences in the pediatric population compared to adults that may modify outcomes from TBI. Created with Biorender.com.

**Figure 3 brainsci-09-00319-f003:**
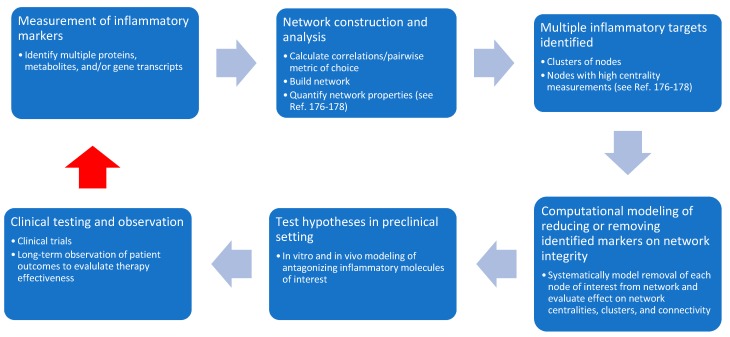
Integrating network analysis into preclinical and clinical research to further our understanding of, and develop novel therapies for, post-TBI inflammation. The red arrow indicates feedback from the clinic to the bench to facilitate building and exploring our knowledge base related to inflammation in pediatric TBIs.

**Table 1 brainsci-09-00319-t001:** Summary of Mayo Clinic classification of traumatic brain injury (TBI) severity, adapted from [7], and Glasgow Coma Scale (GCS) classification for TBI, adapted from [9]. LOC, loss of consciousness; PTA, post-traumatic amnesia; GCS, Glasgow Coma Scale; ICH, intracranial hemorrhage; SAH, subarachnoid hemorrhage.

Severity	Clinical Criteria	Glasgow Coma Scale (GCS) Score
Symptomatic (Possible)	- Blurred Vision - Confusion - Dazed - Dizziness - Focal neurologic symptoms - Headache - Nausea	Mild: 13–15
Mild (Probable)	- LOC < 30 min - Post-traumatic anterograde amnesia < 24 h - Depressed, basilar, or linear skull fracture (dura intact)	
Moderate–Severe (Definite)	- LOC > 30 min - PTA ≥ 24 h - GCS < 13 - One or more of ICH, subdural/epidural hematoma, cerebral contusion, hemorrhagic contusion, penetrating TBI, SAH, brain stem injury	Moderate: 9–12 Severe: 3–8

**Table 2 brainsci-09-00319-t002:** Overview of secondary mechanisms of injury and their connection to inflammation.

Biological Mechanism	Connection to Inflammation
Excitotoxicity	- As glutamate is known to be a co-stimulator of T cells and a potent gliotransmitter, decreased uptake of glutamate, via downregulation of excitatory amino acid transporters on astrocytes [39] and alterations of GABAergic interneurons [40,41], reduces inhibition of neighbouring excitatory circuits and can activate immune cells (astrocytes, microglia, etc.)
Mitochondrial Dysfunction & Metabolic Disruption	- Increased Ca^2+^ influx can overload the mitochondria, promoting network fission [42]. Mitochondrial network fission increases reactive oxygen species (ROS) production, reduces oxidative phosphorylation [43], and has been shown to be required for the activation of microglia in vitro [44] - A shift to glycolysis within neurons is enhanced by cytokines produced by nearby immune cells, notably through activation of the PI3K-mTOR pathway [45]. In combination with mitochondrial dysfunction and increased ROS, this can lead to neuronal degeneration and increase levels of damage-associated molecular patterns (DAMPs) in the surrounding tissue
Increased Oxidative Stress	- Stabilizes HIF1α [46] and promotes NLRP3 [47] inflammasome formation necessary for the production of inflammatory mediators
Weakened BBB Integrity	- Increased leakage of DAMPs, including GFAP, NFL, p-tau, and UCH-L1, into the bloodstream and extravasation of peripheral immune cells into the brain [48]
Cytoskeletal Breakdown/Protein Aggregation	- Increased Ca^2+^ influx can activate Ca^2+^-dependent enzymes, such as calpains [49], leading to cytoskeletal breakdown. These broken down proteins can then leak into the peripheral circulation and/or aggregate into plaques within the CNS, leading to further inflammation
Cerebral Blood Flow Dysregulation	- Hypoxia or ischemia can kill cells, causing them to release their internal contents and activate surrounding immune cells via DAMPs and PRRs - Hypoxia/ischemia can stabilize HIF1α in an ROS-independent manner [50] to increase production of cytokines
Edema (Vasogenic)	- Facilitated by neurogenic inflammation (release of Substance P and neurokinins) - Increases ability of immune cells to extravasate into the brain - Increases transmission of DAMPs from the brain into the blood to recruit peripheral immune cells
Edema (Cytotoxic)	- Influx of water into the cell can lead to swelling and membrane and organelle disruption, leading to cell death and release of DAMPs into the extracellular space
Glial Cell Activation	- Injury to the CNS activates astrocytes and microglia, which reciprocally signal to activate (and de-activate) gliosis. These signals include an initial burst of purinergic substrates, such as ATP from astrocytes, which activate the P2Y12R and P2X4R purinergic receptors [51,52], leading to microglial process extension towards the injury site - Microglia signal to astrocytes to convert them to a neuroprotective phenotype via downregulation of the P2Y1R receptor on the astrocyte surface using TNFα, IL-1β, and IL-6 [53]. The converse can also occur, with microglia inducing a toxic astrocyte phenotype through secreting TNFα, IL-1β, and C1q [54] - CX3CR1 on microglia exhibits a time dependent effect on outcome after injury, playing a key role in inflammation (accumulation of leukocytes [55]), but is required for proper recovery in severe [56] and mild [57] TBI - Formation of a glial scar using Eph/ephrin signaling, namely EphA4 and CSPGs in the CNS [58], can affect vascular permeability and enhance immune cell migration into the injured CNS

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
