# Peer review of "Neuro-Inflammation in Pediatric Traumatic Brain Injury—from Mechanisms to Inflammatory Networks"

_brainsci, 2019, doi:10.3390/brainsci9110319_

Round 1

Reviewer 1 Report

Comprehensive review concerning actual knowledge of neuro-inflammation in pediatric brain injury. Moreover, the paper suggests future directions of research including network modeling. The publication is well referenced and clearly shows limits of translating existing research results e.g. from animal studies to pediatric head injuries.

Minor comment

For readers who ar not too familiar with molecular biology a glossary with a short explanation of mechanisms of action of the different molecules described in the paper would be quite helpful

Reviewer 2 Report

This manuscript provides a review of neuroinflammation after TBI, with a focus on components of the cellular immune response. The authors also include a case for moving away from isolated analysis of single inflammatory mediators and pathways, to a systems biology, network approach to drive the field forward. Key gaps in knowledge in the field - specifically, a lack of data regarding pediatric TBI, female sex, models incorporating rotational injury mechanisms, and chronic time points. 

My primary concern is regarding the pitch and focus of the review. The manuscript (as summarized above) is incompletely/inaccurately described by both the abstract and title, which indicate that the review is focused on neuroinflammation in pediatric TBI. In fact, few studies regarding neuroinflammation in TBI are cited. E.g. several studies on neutrophils in the developing TBI brain from Noble-Haeusslein are notably absent - see - https://www.ncbi.nlm.nih.gov/pubmed/30876905 ; https://www.ncbi.nlm.nih.gov/pubmed/16554253. I suggest either (a) altering the abstract and title to better reflect the manuscript content, including the focus on a systems/network approach; or (b) revise the manuscript to provide a heavier focus on neuroinflammation in the context of pediatric TBI. 

Other suggested edits:

A introduction to the manuscript is missing, but should be included. This will further help to focus the manuscript and guide the reader about what to expect within.  Why was the Mayo Clinic classification used, compared to the GCS, to describe injury severity? The latter is more commonly used and should be detailed. Table 2 appears to be inappropriately placed, very early in the manuscript prior to any introduction of some of the key concepts and terms contained within (e.g. DAMPs, glial cell activation, etc). This would be better positioned later, e.g. after section 3.  Figure 2 - is this an original? If not, please cite the source an ensure permission has been granted for reprint. If it is an original, consider more details to be included in the figure legend (e.g. move text box into figure legend) and annotated figure itself to clearly define the cell types depicted.  References to all tables and figures should be within the preceding text (some references are absent).  Table 3 is extraneous information of indirect relevance to the manuscript's focus - I suggest removing it completely.

Round 2

Reviewer 2 Report

The authors have addressed my concerns.